# Mitigating Object Hallucination via Concentric Causal Attention

**Yun Xing**[1][*]  **Yiheng Li**[1][*]  **Ivan Laptev**[2]  **Shijian Lu**[1][†]
[1] Nanyang Technological University    [2] MBZUAI
https://github.com/xing0047/cca-llava.git

## Abstract

Recent Large Vision Language Models (LVLMs) present remarkable zero-shot conversational and reasoning capabilities given multimodal queries. Nevertheless, they suffer from object hallucination, a phenomenon where LVLMs are prone to generate textual responses not factually aligned with image inputs. Our pilot study reveals that object hallucination is closely tied with Rotary Position Encoding (RoPE), a widely adopted positional dependency modeling design in existing LVLMs. Due to the long-term decay in RoPE, LVLMs tend to hallucinate more when relevant visual cues are distant from instruction tokens in the multimodal input sequence. Additionally, we observe a similar effect when reversing the sequential order of visual tokens during multimodal alignment. Our tests indicate that long-term decay in RoPE poses challenges to LVLMs while capturing visual-instruction interactions across long distances. We propose Concentric Causal Attention (CCA), a simple yet effective positional alignment strategy that mitigates the impact of RoPE long-term decay in LVLMs by naturally reducing relative distance between visual and instruction tokens. With CCA, visual tokens can better interact with instruction tokens, thereby enhancing model's perception capability and alleviating object hallucination. Without bells and whistles, our positional alignment method surpasses existing hallucination mitigation strategies by large margins on multiple object hallucination benchmarks.

## 1 Introduction

Large Vision-Language Models (LVLMs) [46, 45, 84, 71, 6, 15, 5] have drawn increasing attention from the AI research community due to their impressive power in understanding the visual world and unprecedented ability to interact with humans via conversations. Their capability to process multimodal sequences has opened up new possibilities for a wide range of vision and language tasks [32, 2], such as handling interleaved image-text inputs [4, 35] and interactive user queries [82]. However, existing LVLMs still suffer from object hallucination [57, 41, 44, 14], a tendency to generate inaccurate responses that are not factually aligned with image inputs. Such phenomenon challenges the faithfulness and reliability of LVLMs in practical use, impeding their deployments to real-world applications [14].

A wide range of approaches have been proposed to mitigate object hallucination in LVLMs. One straightforward approach involves post-hoc correction using revisor models [73, 83], reducing occurrences of hallucinated responses. Another viable approach is to improve supervised fine-tuning by diversifying instruction tuning data [43] or additionally aligning model responses with human preference [62, 76]. Despite their effectiveness in mitigating LVLM object hallucination, acquiring high-quality annotations can be labor-intensive, making these approaches costly to implement.

---

[*]Equal contribution
[†]Corresponding author

38th Conference on Neural Information Processing Systems (NeurIPS 2024).

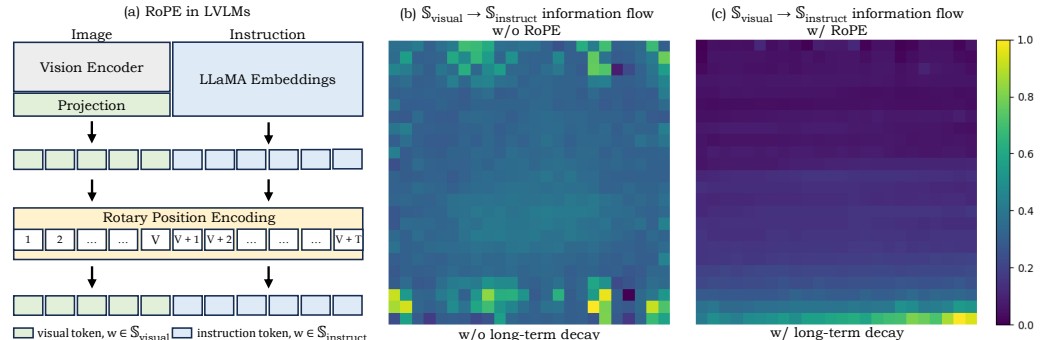

Figure 1: **Long-term decay of RoPE [61] in Large Vision Language Models (LVLMs)**. (a) a schematic view of inference in LVLMs, typically involving a pre-trained vision encoder, a large language model and a projector to map visual tokens to textual space. For each of $V$ visual tokens $\mathbb{S}_{vision}$, we aggregate its information flow to instruction tokens $\mathbb{S}_{instruct}$ and reshape the aggregation results to 2-D ($\sqrt{V}$ by $\sqrt{V}$). Applying RoPE on visual tokens introduces long-term decay as illustrated in (c), referring to the phenomenon where information flowing from visual tokens to instruction tokens gradually decays from lower-right region (rightmost visual tokens in the 1-D sequence) to upper-left region (leftmost visual tokens). For instruction tokens, they have much less direct interaction with leftmost visual tokens as compared with rightmost visual tokens, leading to inferior multimodal alignment in the trained LVLMs. (b) and (c) are derived from the adversarial subset of the $3k$ POPE [41] image-instruction pairs. Best viewed in color.

Recently, several studies explore training-free mitigation of object hallucination by rectifying fallacies in LVLM autoregressive decoding [26, 34]. However, the need to compare among many candidates inevitably slows down the decoding process, making these approaches less efficient during inference.

Distinct from previous efforts, we attend to Rotary Position Encoding (RoPE) [61], a widely used positional dependency modeling design in LVLMs [46, 84], and investigate how it may affect object hallucination in LVLMs. Similar to sinusoidal function [65], RoPE is proposed to encode position information into representations, enhancing model's ability in understanding sequential order of input tokens. In spite of its success in modeling natural language [53, 63, 64], this design leads to long-term decay [61] in multimodal alignment, a phenomenon where information flow from visual tokens to instruction tokens[1] gradually diminishes with increasing relative distance.

We analyze the impact of long-term decay [61, 53] on LVLMs. For every visual token in a multimodal sequence, we aggregate its information flow to all instruction tokens and examine how these aggregations distribute across all visual tokens. As presented, in contrast to information flows of visual tokens without RoPE (Fig. 1, (b)), applying RoPE attenuates information flows of leftmost visual tokens, which are located the farthest from instruction tokens in the sequence (Fig. 1, (c)). Such long-term decay benefits natural language modeling [61], but induces insufficient interactions between visual tokens and instruction tokens, leading to inferior multimodal alignment and object hallucinations in the trained LVLMs (see our experiments in Sec. 3 for details).

To this end, we propose Concentric Causal Attention (CCA), a novel position alignment method for training LVLMs with mitigated object hallucination. CCA consists of a position reorganization module for visual tokens and an accompanying causal mask rectification module for modeling 2-D continuous positional dependency. Instead of following raster-scan [2] sequential order of existing LVLMs, CCA starts from peripheral of 2-D images and ends in centers. Such position alignment strategy enjoys two merits: 1) relative distance from instruction tokens to visual tokens are significantly reduced, alleviating limitations brought by long-term decay in RoPE; 2) rectified causal attentions follow 2-D spatial locality of images, as compared to 1-D causal attention originally designed for natural languages. We carry out pre-training and instruction tuning as [46] and verify our trained model on multiple object hallucination benchmarks [41, 57, 20] (+4.24% on Accuracy and +2.73% on F1 score, as compared to the state-of-the-art method [34] on POPE). From a broader perspec-

---

[1]Information flow here refers to self-attentions from instruction tokens to visual tokens.

[2]2-D image tokens are flattened from left to right, top to bottom, into 1-D visual token sequence.

tive, our method also improves general perception capability of LVLMs. Preliminary experiments show that our positional alignment approach surpasses the baseline consistently over 6 multimodal benchmarks [36, 48, 22, 28, 49, 8].

Our contributions are three-fold. First, we perform in-depth analysis on correlation between rotary position encoding and object hallucination in large vision-language models. Second, motivated by our analysis, we propose Concentric Causal Attention (CCA), a simple yet effective method to mitigate LVLM object hallucination caused by RoPE long-term decay. Third, experiments on multiple benchmarks and comparisons with the state-of-the-art methods support efficacy of our design.

## 2 Related Works

**Large Vision Language Models**. Language modeling has made notable progress in recent years, evolving from robust representation models [17, 56, 55] to instruction-tuned conversational chat-bots [63, 64, 12, 1]. These achievements have driven research in creating Large Vision Language Models (LVLMs) that can manage multimodal inputs [72, 46, 45, 84, 71, 6, 67, 40, 51, 39]. Pioneering studies in this field [2, 4, 38, 37] connect a vision-only encoder with a powerful frozen language-only model to bridge modality gap, enabling dense interactions across visual and textual features. Powered by instruction-tuned LLMs [12], LLaVA [46], InstructBLIP [15] and MiniGPT4 [84] allow interactive conversations between trained models and users. On top of these studies, LVLMs are empowered with more advanced capabilities, such as engaging in referential dialogues [7, 74, 81, 54, 77], handling interleaved image-text data [2, 4, 35] or understanding visual prompts, like point or box inputs from users [54, 82, 9, 77]. Despite advancements in LVLMs, many of these models still generate inaccurate responses not aligned with visual inputs.

**Object Hallucination** refers to a common problem of existing LVLMs [14, 44, 21, 41, 68, 52, 3, 19, 66]. Specifically, LVLMs tend to generate inaccurate responses that are not factually aligned with image inputs. To address this issue, several recent explorations [73, 83, 33] resort to post-hoc correction of model hallucinated outputs. These methods rely on either external models [47] to correct hallucinated responses [73] or on self-correction techniques [33, 70]. However, both of these methods break end-to-end inference scheme. In contrast, [43, 76, 62, 29, 78, 75] ground their approaches on improving instruction tuning, by either diversifying instruction data or aligning model responses with human feedback. However, acquisition of more instruction data or preference data is labor-intensive. Recently, several studies attempt to mitigate object hallucination in a training-free manner [26, 34, 10]. However, the need to compare among many candidates inevitably slows down the decoding process, making these approaches less efficient during inference. From a distinct perspective, we ground our design in correlation between widely adopted rotary position encoding and object hallucination.

**Position Encoding in Transformers**. Transformer models [65] do not inherently comprehend sequential information of input tokens, which is inferior for modeling sequential data like natural language as compared to recurrent structures like [24]. To mitigate this issue, [65] introduces sinusoidal position encodings to incorporate position information to input embeddings. In addition, several studies resort to learnable position encodings [18], which allow their models to update positional parameters during training. In contrast to absolute position encodings, relative position encodings [59, 31, 23, 27] focus on relative position among tokens. They integrate position information in self-attentions, presenting potential for modeling sequences with variable lengths [61, 53]. Among these studies, Rotary Position Encoding (RoPE) [61] encodes position information by multiplying input embeddings with rotation matrices. In comparison to other position encoding designs, RoPE is capable of equipping linear self-attention with relative position encoding, which is proven effective for pre-training large language models [63, 64]. A few recent studies explores RoPE for vision tasks [13, 50, 69], showcasing its potential to domains beyond natural language. In this paper, we investigate the role of RoPE in LVLMs and how it affects object hallucination in these models.

## 3 Motivation

In this section, we further examine the long-term decay in RoPE and conduct quantitative analyses to illustrate its correlation with object hallucination. We begin with a brief introduction to the widely adopted LVLM architecture and how RoPE [61] is applied in LVLMs. Then, we highlight the long-term decay in RoPE [61, 53], which benefits language modeling but is under-explored for

multimodal alignment. Finally, we examine the role of RoPE in LVLM object hallucination through comparative experiments, which forms a strong foundation of our design.

**LVLM**. Typically, an LVLM $\mathcal{F}$ is composed of a pretrained vision encoder $\mathcal{F}_v$, a large language model $\mathcal{F}_t$ and a projector module $f$ that maps visual embeddings to textual space. Given an image input $I_v$ and instruction input $I_t$ (e.g., *"please describe this image in detail"*), $\mathcal{F}$ encodes these two inputs into a multimodal sequence $\mathbb{S} = \{\mathbb{S}_{vision}, \mathbb{S}_{instruct}\}$, where $\mathbb{S}_{vision} = f(\mathcal{F}_v(I_v)) = \{w_m\}_{m=1}^{V}$ and $\mathbb{S}_{instruct} = \mathcal{F}_t(I_t) = \{w_m\}_{m=1}^{T}$ represent visual and instruction tokens of lengths $V$ and $T$, respectively. In such sequence, visual and instruction tokens share the same dimension $d$, noted as $w_m \in \mathbb{R}^d$.

**Rotary Position Encoding in LVLM**. In LLMs like LLaMA [63] and its multimodal successors, RoPE [61] encodes position information with input tokens by multiplying every token $w_m$ with a rotation matrix $R_{\theta,m}^d$,

$$R_{\theta,m}^d = \begin{pmatrix} \cos m\theta_1 & -\sin m\theta_1 & 0 & 0 & \cdots & 0 & 0 \\ \sin m\theta_1 & \cos m\theta_1 & 0 & 0 & \cdots & 0 & 0 \\ 0 & 0 & \cos m\theta_2 & -\sin m\theta_2 & \cdots & 0 & 0 \\ 0 & 0 & \sin m\theta_2 & \cos m\theta_2 & \cdots & 0 & 0 \\ \vdots & \vdots & \vdots & \vdots & \ddots & \vdots & \vdots \\ 0 & 0 & 0 & 0 & \cdots & \cos m\theta_{d/2} & -\sin m\theta_{d/2} \\ 0 & 0 & 0 & 0 & \cdots & \sin m\theta_{d/2} & \cos m\theta_{d/2} \end{pmatrix} \tag{1}$$

where $m \in [1, ..., V + T]$ indicates position of input token $w_m$ and $\{\theta_i = 10000^{-2(i-1)/d}\}, i \in [1, 2, ..., d/2])$ are pre-defined sinusoidal function values following [65]. In LVLMs like LLaVA [46], rotary matrices $R_{\theta,m}^d$ are applied to query and key tokens in all decoder layers, such that relative position dependency among tokens are modeled and integrated in self-attentions across the network. In comparison to absolute position encodings [65] and learnable position encodings in ViT [18], RoPE captures relative distance among input tokens and has the potential to extend the input context window beyond a fixed length [53].

**RoPE Long-term Decay**. Assume a query token $q_i$ at position $i$ and a key token $k_j$ at position $j$, which are derived from input tokens $w_i$, $w_j$. The self attention $a_{i,j}$ between tokens $q_i$ and $k_j$ can be calculated via

$$\mathrm{a}_{i,j} = \mathrm{softmax}(\frac{q_i^T \cdot k_j}{\sqrt{d}}) \tag{2}$$

RoPE applies rotation matrix $R_{\theta,m}^d$ to the self-attention above, which is in the form of,

$$\mathrm{a}_{i,j} = \mathrm{softmax}(\frac{q_i^T \cdot (R_{\theta,i}^d)^T \cdot R_{\theta,j}^d \cdot k_j}{\sqrt{d}}) = \mathrm{softmax}(\frac{q_i^T \cdot R_{\theta,j-i}^d \cdot k_j}{\sqrt{d}}) \tag{3}$$

where $j - i$ stands for relative position between $q_i$ and $k_j$. The long-term decay refers to the decrease of $\mathrm{a}_{i,j}$ as the relative distance $j - i$ increases. As presented in Fig. 1 (c), visual-to-instruction information flow (i.e., instruction-to-visual self-attention) is less significant when $j - i$ is large and vice versa.

This is favorable for pre-trained LLMs like LLaMA [63], as it aligns with language modeling intuition: pairs of tokens with a long relative distance should have weaker connection. However, we observe that this property brings negative effect in multimodal alignment, in which case visual tokens far from instructions are less attended. This is not expected for multimodal alignment, as the connection between instruction tokens and visual tokens should not be attenuated by their relative distances.

**Pilot Experiment**. We quantitatively examine the effect of RoPE long-term decay on LVLM object hallucination. To determine how object hallucination is influenced by the distance between visual and instruction tokens, we first train two LVLMs [3] following [46] with two different position alignment strategies, including:

---

[3]Training details for these two models are in Appendix C.1.

(a) Aggregated correct responses with $\mathcal{F}^b$ baseline raster scan    (b) Aggregated correct responses with $\mathcal{F}^r$ reverse raster scan

Figure 2: **Motivation Experiment.** Given an image $I_v$ with object $O_v$, we crop $O_v$ and paste it to various spatial positions $\{v_1, ..., v_k\}$ within a pre-defined template. For every pasting position, we ask two LVLMs ($\mathcal{F}_b$ and $\mathcal{F}_r$) if object $O_v$ is in this template, where $\mathcal{F}_b$ refers to a baseline model that follows raster-scan positional alignment strategy and $\mathcal{F}_r$ refers to a model that resorts to reversal raster-scan position alignment strategy. The total number of correct responses at different pasting positions $\{v_1, ..., v_k\}$ is reported in (a) and (b), which refers to results from model $\mathcal{F}_b$ and $\mathcal{F}_r$, respectively. We observe that LVLM $\mathcal{F}_b$ are more likely to generate correct responses when pasting object $O_v$ to lower region, while $\mathcal{F}_r$ are less hallucinated when pasting object $O_v$ to upper region. Pasting positions with the most and the least correct responses are highlighted in solid-line and dotted-line red boxes. More details are provided in Appendix C.1. Best viewed in color.

- $\mathcal{F}^b$ (*raster-scan*): it follows [46] the position alignment strategy on visual tokens $\mathbb{S}_{vision}$. Under this scenario, visual tokens follow a sequential order, starting from upper-left corner to lower-right corner of input 2-D visual features, row by row. The order of a multimodal sequence $\mathbb{S}$ is in format of $\{1, 2, ..., V, V + 1, ..., V + T\}$.[4]

- $\mathcal{F}^r$ (*reverse raster-scan*): it reverses the sequential order of visual tokens $\mathbb{S}_{vision}$. In this case, sequence order of visual tokens starts from lower-right corner of input 2-D visual features to upper-left corner, row by row. The order of full multimodal sequence $\mathbb{S}$ is in format of $\{V, V - 1, ..., 1, V + 1, ..., V + T\}$.

The *reverse raster-scan* model $\mathcal{F}^r$ alters relative positions between visual tokens $\mathbb{S}_{vision}$ and instruction tokens $\mathbb{S}_{instruct}$. For example, for instruction token $w_{V+1}$, its relative distance to visual token $w_V$ changes from 1 to $V$, resulting in weaker correlations between $w_V$ and $w_{V+1}$.

Our experiment setup is as follows. Given an image $I_v$, we follow [41] and ask questions in a polling-base manner. Specifically, for an object $O_v$ in image $I_v$, we follow the instruction format of "is there a/an {object} in this image?" to test our models. We crop region of object $O_v$ from $I_v$ according to its bounding box annotation and paste the cropped object over different positions of a pre-defined image template (more details are covered in Appendix C.1). This results in new images $\{I_{v_1}, ..., I_{v_k}\}$, where $\{v_1, ..., v_k\}$ indicates different pasting positions. We carry out these testing over $N$ images from [42] and aggregate correct responses with respect to pasting positions $\{v_1, ..., v_k\}$.

**RoPE affects object hallucination**. The quantitative results of model $\mathcal{F}^b$ and $\mathcal{F}^r$ are visualized in Fig. 2 (a) and (b), respectively. For model $\mathcal{F}^b$, we find that the response is less likely correct when object $O_v$ is pasted on the upper part of the image, and it is more likely correct when object $O_v$ is pasted on the lower part of image template. This is in stark contrast to $\mathcal{F}^r$ experimental results: model responses are more likely to be correct when pasting image crop $O_v$ on the upper part of images, while less likely to be correct when pasting position is the lower part. For model $\mathcal{F}^r$, we note

---

[4]For demonstration purpose, we assume visual tokens are pre-pended before instruction tokens. For implementation, we adapt our design for flexible structure of multimodal sequences.

that visual tokens of lower part is far from instruction tokens in relative distance, corresponding to worse performance in object hallucination. We can thus conclude that RoPE long-term decay affects object hallucination for LVLMs, which requires special care to mitigate this issue.

# 4 Concentric Causal Attention

To this end, we introduce Concentric Causal Attention, a simple position alignment strategy that mitigates object hallucination by tackling the long-term decay issue originated from RoPE. Our methodology is guided by two key principles,

- Alleviate the effect of long term decay on object hallucination by minimising overall relative distance between visual tokens $\mathbb{S}_{vision}$ and instruction tokens $\mathbb{S}_{instruct}$.
- Mitigate performance discrepancy between *raster scan* model $\mathcal{F}^b$ and *reverse raster scan* model $\mathcal{F}^r$.

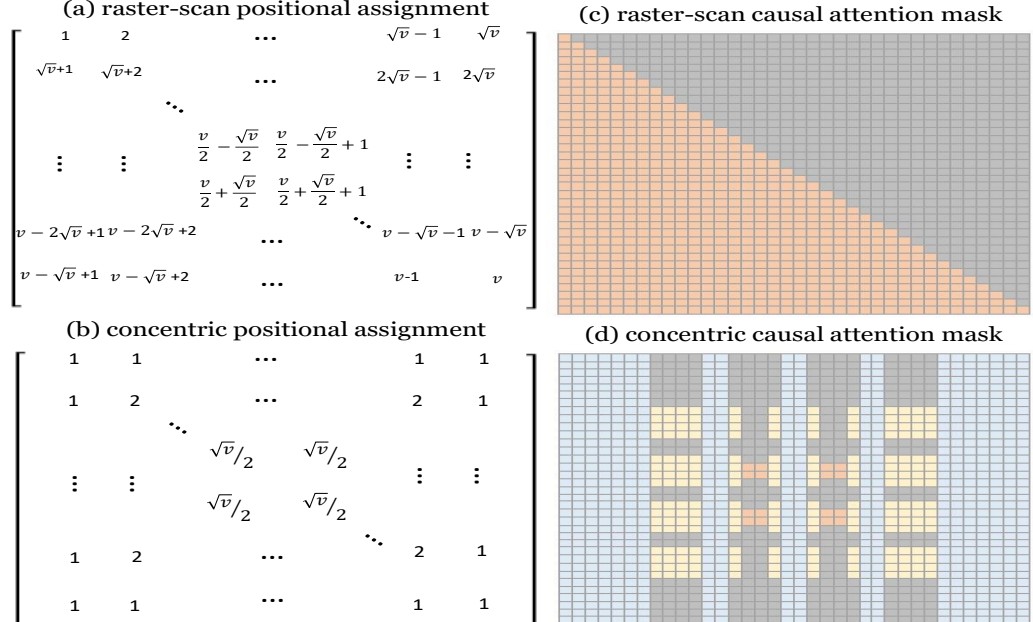

Figure 3: An overview for Concentric Causal Attention. **Left: Visual Token Re-organization**. In comparison to *raster-scan positional alignment* in (a), we design *concentric position alignment* in (b) which shortens visual-instruction distance and retains spatial locality for 2-D data like images. **Right: Concentric Causal Masking**. By default as in (c), a visual token attends to all preceding visual tokens in a 1-D sequence. In contrast, our *concentric causal attention* in (d) models 2-D continuous positional dependencies among visual tokens, where center visual tokens attend to peripheral ones. Causal masks are $V$ by $V$ where in this case $V$ is 36 for demonstration purpose. Best viewed in color.

**Concentric Positions**. In existing LVLMs such as LLaVA [46], visual tokens are perceived in 1-D continuous sequence (raster-scan position alignment as illustrated in Fig. 3 (a)) and concatenated with instruction tokens for multimodal alignment. We note that such row-by-row positional alignment strategy is not natural for 2-D image data, as it breaks spatial continuity on column dimension. Due to the long-term decay in RoPE, information flow from visual token $w_m$ to $w_{m+1}$ differs from that to $w_{m+\sqrt{V}}$, which diverges from spatial locality of 2-D visual features.

Instead of adopting raster-scan sequential order, we design a concentric positional alignment strategy as illustrated in Fig. 3 (b). In our design, position $m$ of visual tokens are organized in a form of 2D concentric square, which increases from the peripheral of 2-D inputs to the center. In comparison to sequence order of $\{1, 2, ..., V\}$ for visual tokens $\mathbb{S}_{vision}$, such concentric positional alignment reduces relative distance between visual and instruction tokens $\mathbb{S}_{instruct}$. For a visual token sequence

of length $V$ and a instruction token sequence of length $T$, the maximum distance between visual tokens $\mathbb{S}_{vision}$ and instruction tokens $\mathbb{S}_{instruct}$ is $(\frac{\sqrt{V}}{2} + T - 1)$. This concentric sequential ordering also better maintains 2-D spatial locality of visual tokens. Under this scenario, visual tokens that are closer in euclidean distances are causally correlated when position $m$ increases. Meanwhile, visual tokens that share the same position are correlated in visual self-attention. We note that such design mitigates negative effect from RoPE long-term decay, via decreasing relative distances between $\mathbb{S}_{vision}$ and $\mathbb{S}_{instruct}$ while keeping causal inference scheme in pre-trained LLMs like LLaMA [64].

**Concentric Causal Masking**. Another part of our method resorts to modification of default causal attention masking towards our concentric visual token reorganization. As presented in Fig. 3 (c), a query feature $q_m$ (derived from $w_m$) only attends to preceding key features $k_{<=m}$. Likewise for our method, we follow the same principle to force causal attention masking in 2-D visual inputs. We visualize our masking in Fig. 3 (d), where the total length of visual tokens are 36 (6 by 6). Combining visual token re-organization with concentric causal masking, our method models 2-D continuity for visual inputs and effectively mitigates the object hallucination issue brought by long-term decay in RoPE.

## 5   Experiments

We first describe training details for our position alignment approach and evaluation setups in Sec. 5.1. Subsequently, we report results for several popular benchmarks that demonstrates efficacy of our simple design in the remaining subsections. Further, we present qualitative comparison in Appendix D.2 where our approach generates less hallucinated responses. From a broader scope, we present that our positional alignment strategy benefits general perception capability of LVLMs, where preliminary experiments show that it surpasses the baseline consistently over six multimodal benchmarks [36, 28, 22, 48, 8, 49]. We refer to these results in Appendix D.1 due to page limits. By default, we conduct our training and evaluation with Vicuna-7B [11] model, unless otherwise stated.

### 5.1   Training Details

Following [46, 45], we adopt pre-trained CLIP ViT-L/14 [55] with 336x336 resolutions as visual encoder and Vicuna-7B [12] as LLM, and a 2-layer MLP that connects the visual encoder and LLM. Training consists of two stages, including 1) a pre-training over CC-558K dataset [46] with global batch size of 256 and 2) a instruction tuning with a 665k multi-turn conversation dataset [45] with global batch size of 128.

### 5.2   POPE

Polling-based Object Probing Evaluation (POPE) [41] is proposed to provide a detailed evaluation of object hallucination in LVLMs, by querying the models about presence of specific objects in given images with yes-or-no questions. POPE adopts three sampling options to sample negative objects: random, popular and adversarial. We refer to [41] for these setups. Following [34], three datasets are included in our evaluation, including COCO [42], GQA [28] and A-OKVQA [58]. For each evaluation setup, every subset includes 3,000 questions for 500 images, which leads to 27,000 yes-or-no questions in total.

The experimental results are presented in Tab. 1. Our method achieves the highest accuracy and F1 scores across all datasets and negative sampling setups. By re-organization of visual tokens and concentric masking, our approach achieves 5.48%, 7.86% and 6.70% accuracy improvement and 5.89%, 7.71% and 6.19% F1 score improvement over the baseline model [46]. We also observe consistent and notable performance gains against state-of-the-art hallucination mitigation methods. CCA surpasses VCD [34] by 1.02%, 4.51% and 2.65% on three datasets. Particularly, we observe 3.09%, 5.01% and 3.59% F1 score improvement over adversarial evaluation set, which selects the most frequent co-occuring objects with ground-truth objects in image inputs, posing challenges for LVLMs to discern spurious correlation. Our trained model is also comparable to LLaVA-RLHF model (with Vicuna-13B as its LLM) [62] that additionally aligns model responses with human preference. These results indicate importance of re-organizing visual tokens in vision-language alignment.

Table 1: **POPE Results**. acc: accuracy. f1: f1 score, measured by precision and recall. Baseline and VCD results are reported by paper [34].

| Evaluation | Method | random | | popular | | adversarial | | average | |
|---|---|---|---|---|---|---|---|---|---|
| | | acc | f1 | acc | f1 | acc | f1 | acc | f1 |
| MSCOCO [42] | baseline | 83.29 | 81.33 | 81.88 | 80.06 | 78.96 | 77.57 | 81.38 | 79.65 |
| | VCD [34] | 87.73 | **87.16** | 85.38 | 85.06 | 80.88 | 81.33 | 84.66 | 84.52 |
| | LLaVA-RLHF [62] | 85.90 | 83.92 | 83.90 | 82.05 | 82.60 | 80.88 | 84.13 | 82.28 |
| | CCA-LLaVA | **88.03** | 86.65 | **86.87** | **85.54** | **85.67** | **84.42** | **86.86** | **85.54** |
| A-OKVQA [58] | baseline | 83.45 | 82.56 | 79.90 | 79.59 | 74.04 | 75.15 | 79.13 | 79.10 |
| | VCD [34] | 86.15 | 86.34 | 81.85 | 82.82 | 74.97 | 77.73 | 80.99 | 82.30 |
| | LLaVA-RLHF [62] | 87.67 | 86.60 | 85.20 | 84.34 | 79.97 | 79.92 | 84.28 | 83.62 |
| | CCA-LLaVA | **90.27** | **89.71** | **88.40** | **87.98** | **82.30** | **82.74** | **86.99** | **86.81** |
| GQA [28] | baseline | 83.73 | 82.95 | 78.17 | 78.37 | 75.08 | 76.06 | 78.99 | 79.13 |
| | VCD [34] | 86.65 | 86.99 | 80.73 | 82.24 | 76.09 | 78.78 | 81.16 | 82.67 |
| | LLaVA-RLHF [62] | 84.93 | 83.38 | 81.37 | 80.23 | 78.30 | 77.70 | 81.53 | 80.44 |
| | CCA-LLaVA | **88.40** | **87.68** | **86.47** | **85.91** | **82.20** | **82.37** | **85.69** | **85.32** |

## 5.3 CHAIR

We further evaluate our method on Caption Hallucination Assessment with Image Relevance (CHAIR) metric. CHAIR was a pioneering study introduced to measure object hallucination in image captioning [57]. It quantifies the factuality of a model by calculating the proportion of objects not present in ground truth over all objects in caption output. It contains both instance level score $CHAIR_I$ (shorted for $C_I$) and sentence level score $CHAIR_S$ ($C_S$) which holistically assess a model's performance. Specifically, CHAIR metric is formulated as:

$$C_S = \frac{|\{\text{sentences with hallucinated objects}\}|}{|\{\text{all sentences}\}|}, \ C_I = \frac{|\{\text{hallucinated objects}\}|}{|\{\text{all mentioned objects}\}|}$$

where lower scores corresponds to better performance. Following previous studies [26], we prompt LVLMs with *"Please describe this image in detail."*. Note that LVLM's performance on CHAIR metric is highly dependent on their output sentence length. Short and succinct responses have less chances to make mistakes and thus would generally have better CHAIR scores. Different textual prompts such as *"in detail"* and *"in brief"* also influences output length and creates bias in CHAIR evaluation [41]. To offset the influence of output length and prompt phrasing and ensure fair basis of comparison, we follow the experimental setup in OPERA [26] and set the maximum text token to 64 and 512 respectively to examine hallucination on both short and long responses. Following [26], we sample 500 images from COCO VAL 2014 [42] to generate descriptions from different models and hallucination mitigation methods.

Our image caption evaluation result on CHAIR is shown in Tab. 2. For greedy decoding, our model surpasses baseline model [46] by 3.2% while maintaining high object recall (80.3% v.s. 80.4%) for long-response generation (by setting max new tokens to $512$). Note that longer textual responses suggests more significant distance between visual and instruction tokens, leading to higher hallucination rates [83], which can be improved by our approach that reduces relative distance between visual and textual tokens. Our results are comparable against LLaVA-RLHF [62] over this setup. On short responses, our model also outperforms baseline model by 2.8% on sentence-level and 0.8% on instance-level while maintaining high object recall.

Our approach is also effective when using beam search for autoregressive decoding. We surpass the baseline by 0.8% and 0.5% on long-response generation, and 2.2% and 0.5% on short-response generation for $C_S$ and $C_I$, respectively. Our approach is also complementary to OPERA [26]. In comparison to baseline model that using OPERA decoding, our approach are 1.8% and 1.1% better for $C_S$ and $C_I$ on long-response setting. We observe consistent performance gains in short-response generation (1.6% for $C_S$ and 0.9% for $C_I$). Quantitative evaluations on open-ended generation indicates importance of a better positional alignment strategy and efficacy of our design.

Table 2: **CHAIR results**. For evaluation setups, 512 and 64 refer to a hyperparater that relates to the length of LVLM repsonses, corresponding to long-text and short-text generation, respectively.

| Evaluation | Method | 512 | | | | 64 | | | |
|---|---|---|---|---|---|---|---|---|---|
| | | $C_\downarrow^S$ | $C_\downarrow^I$ | $rec_\uparrow$ | $len$ | $C_\downarrow^S$ | $C_\downarrow^I$ | $rec_\uparrow$ | $len$ |
| *greedy* | baseline | 46.2 | 12.9 | 80.3 | 97.2 | 21.0 | 6.2 | 66.3 | 54.9 |
| | LLaVA-RLHF [62] | 43.6 | **10.5** | 78.0 | 117.9 | 19.6 | 5.4 | 64.9 | 54.0 |
| | CCA-LLaVA | **43.0** | 11.5 | **80.4** | 96.6 | **18.2** | **5.4** | **66.7** | 54.5 |
| *beam* (5) | baseline | 49.4 | 13.9 | 79.9 | 96.1 | 18.2 | 5.8 | 64.0 | 52.7 |
| | OPERA [26] | 46.8 | 13.4 | 79.6 | 93.2 | 17.8 | 5.9 | 64.3 | 53.0 |
| | CCA-LLaVA | 48.6 | 13.4 | **79.9** | 94.2 | **16.0** | 5.3 | 64.8 | 52.7 |
| | CCA-LLaVA + OPERA [26] | 45.0 | **12.3** | 79.5 | 91.8 | 16.2 | **5.0** | **65.0** | 52.9 |

## 5.4 MME

The MME hallucination subset extends scope beyond object hallucination. Following [34], we evaluation 4 perception sub-tasks that examines LVLMs on object-level and attribute-level hallucinations, including measure of object existence, count, position and color. As presented in Tab. 3, our method surpasses the baseline by 76.33 on these tasks. In comparison to previous hallucination mitigation method VCD, our approach demonstrates non-negligible performance gains over all subtasks (e.g., 2.00 improvement from VCD v.s. 24.00 improvement from our method). These results indicate the potential of CCA to improve general perception capability of LVLMs.

Table 3: MME results.

| Model | Object-level | | Attribute-level | | Total |
|---|---|---|---|---|---|
| | existence | count | position | color | |
| baseline | 175.67 | 124.67 | 114.00 | 151.00 | 565.33 |
| OPERA [26] | 180.67 | 133.33 | 123.33 | 155.00 | 592.33 |
| VCD [34] | 184.66 | 138.33 | **128.67** | 153.00 | 604.66 |
| CCA-LLaVA | **190.00** | **148.33** | 128.33 | **175.00** | **641.66** |

Table 4: LLaVA Bench (In-the-Wild) results.

| Model | **Complex** | **Detail** | **Conv** | Overall |
|---|---|---|---|---|
| baseline | 65.8 | 51.2 | 54.6 | 58.9 |
| OPERA [26] | 66.4 | **56.9** | 44.0 | 61.3 |
| VCD [34] | **69.6** | 51.8 | 57.3 | 61.6 |
| CCA-LLaVA | 66.1 | 53.9 | **69.4** | **64.3** |

## 5.5 GPT4V-Aided Evaluation

We also evaluate our approach on LLaVA-Bench (In-the-Wild) [46], composed of 24 images with 60 questions in total. LLaVA-Bench (In-the-Wild) constitutes three types of questions, including conversation, detailed description and complex reasoning. Following [46, 26], we ask these models to generate responses and let the text-only GPT-4 [1] be the judge to rate these responses. The results are presented in Tab. 4. In comparison to OPERA [26] that specializes in open-ended generation, our method still stands out when examined by GPT-4 according to detailness and correctness, suggesting efficacy of our positional alignment strategy on generating accurate long responses.

## 6 Conclusion and Limitations

In this paper, we aim to mitigate object hallucination in Large Vision-Language Model (LVLM). We perform in-depth analysis on correlation between object hallucination and Rotary Position Encoding, a widely used positional dependency modeling design in existing LVLMs. We find that LVLMs are more likely to hallucinate when relevant visual cues are distant from instruction tokens in 1-D multimodal sequence, due to long-term decay in RoPE. To this end, we propose Concentric Causal Attention, a simple yet effective positional alignment strategy that reduces relative distances between visual and instruction tokens, alleviating negative impact brought by RoPE long decay on object hallucination. Experimental results over multiple evaluation benchmarks supports our design, indicating importance of better position alignment strategy.

**Limitation**. While this study shows improvements on mitigating object hallucination in LVLMs, our focus is only limited to handling of image-text inputs. We consider positional alignment strategy for other modalities of input data as future works, such as audio or video inputs that differs from image-text modalities.

## Acknowledgments and Disclosure of Funding

This project is funded by the Ministry of Education Singapore, under Tier-1 project scheme with project number RG18/22 and Tier-2 project scheme with project number MOE-T2EP20220-0003.

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

# Appendix

## A    Broader Impact

Like other LVLMs, models trained by our CCA approach have their potential benefits and risks when they are publicly released. As our approach is validated on LLaVA which constitutes CLIP, Vicuna and LLaMA, our trained models may inherit risks from these pre-trained visual encoders and large language models, including handling malicious inputs, hallucination or potential biases. We mitigate these issues following other LVLMs.

## B    RoPE in LLaMA

We further clarify the role of Rotary Position Encoding (RoPE) [61] in LLaMA architecture with a separate illustration. As Fig. 4 shows, RoPE is densely involved in LLaMA [63, 64], namely in all self-attention layers. This is architecturally distinct from how positions are involved in ViT, where absolute PEs are only added once right after patch embedding layer. As most open-source LVLMs are using LLaMA as their language backbones [46, 84, 15, 67, 30, 79], it is noteworthy to study how RoPE may affect multimodal perception when we connect pretrained vision models (e.g., CLIP) with LLaMA.

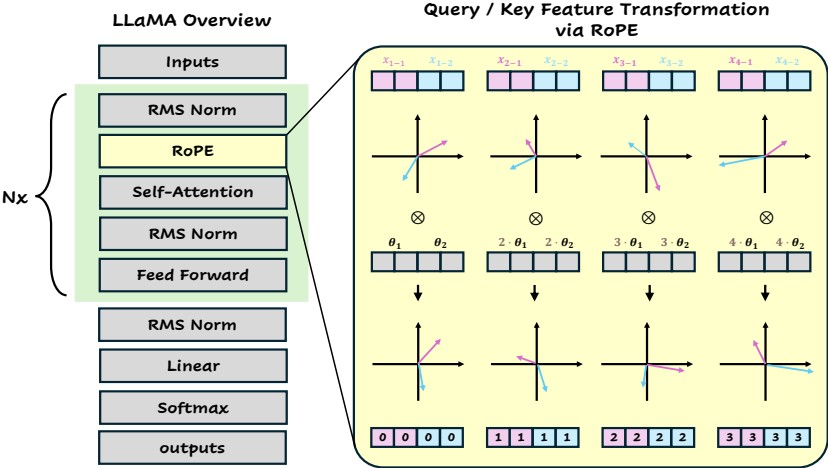

Figure 4: **RoPE in LLaMA**. A schematic view for LLaMA where RoPE is highlighted, and an example illustration on how RoPE is applied over query or key feature. We use a short input sequence with length of 4 and feature dimension of 4 for demonstration purpose. Input tokens are rotated with angles, subject to token positions. For mathematical definition, please refer to Sec. 3.

## C    Implementation Details

We include more details here about implementation for Fig. 1 and Fig. 2 results in main paper, including data and model architecture we use, and training details we follow.

### C.1    Pilot Experiment

**Training**. As described in Sec. 3 of main paper, we train a baseline LVLM $\mathcal{F}_b$ that follows raster-scan positional alignment and another LVLM $\mathcal{F}_r$ that follows reversal raster-scan position alignment. For these two models, we carry out two-stage training following [46], except for the second stage we train both models for 20K steps with LoRA [25] due to resource limitations. Both experiments share the same training hyper-parameters as 665K full schedule training.

**Inference**. We sample 3,000 annotations from COCO VAL 2014 [42] to carry out our motivation experiments. For each annotation with its corresponding image, we crop an object according to its bounding box and paste it within a pre-defined template (a visually gray image), which is initialized with ImageNet [16] average pixel values. We test $k$ spatial positions $\{v_1, ..., v_k\}$, where $k$ is set to $144$, resulting in resolution of $12$ by $12$ for both aggregated results in Fig. 2. Workflow on how we construct such synthetic data is further presented in Fig. 5.

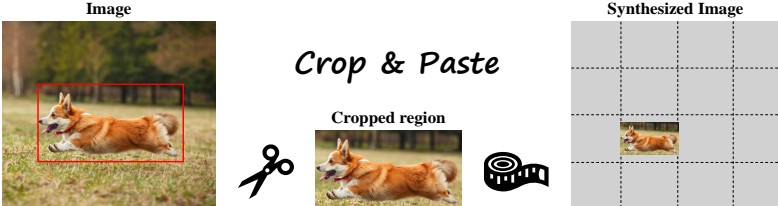

Figure 5: Workflow illustration on how we synthesize testing data. Given an image and box annotation for one object instance, we crop it and paste it on a template image, initialized with ImageNet mean pixel values. We paste every cropped region on every spatial position. Resulting data constitutes a large amount of questions about object existence, diverse in spatial positions.

## C.2 Information Flow

We reveal long-term decay property of RoPE [61] in scope of LVLMs. To implement this, we use 3,000 image-query pairs from POPE [41] adversarial setup, and LLaVA-1.5-7B [46] as our LVLM. For each image-query pair, we extract and aggregate self-attentions from the first decoder layer of LLaMA [64]. We average obtained self-attentions across heads and images to obtain our quantitative results in Fig. 1. A pseudo code is provided below for further clarification.

```python
def compute_vis_inst_flow(
    attn,
    img_token_pos,
    img_token_len
):
    """
    Return
        information flow from visual (vis) to instruction (inst) tokens.

    Input
        attn - self attentions.
        img_token_pos - where image sequence starts.
        img_token_len - sequence length for visual tokens.
    """
    inst_vis_attn = attn[
        :,
        :,
        img_token_pos + img_token_len + 1:,
        img_token_pos: img_token_pos + img_token_len
    ]
    # average across images, heads, and instruction tokens.
    vis_inst_flow = inst_vis_attn.mean(dim=(0, 1, 2))
    return vis_inst_flow
```

# D   More Results

## D.1   Comparison over Multiple-Choice Benchmarks

Beyond the scope of visual hallucination, we consider our proposed positional alignment as a general approach for improving perception capability for LVLMs. We further evaluate our trained model over six benchmarks that examines LVLMs general perception capability, including SEED-Bench [36], ScienceQA [49], GQA [28], Vizwiz [22], MMBench [48] and MMStar [8] which evaluates LVLMs perception capability with multiple choice questions. We use lmms-eval [80] to do our comparison.

For details of our evaluation benchmarks, SEED-Bench [36] consists of 19k multiple choice questions with human annotations, while spanning 12 evaluation dimensions, including both image and video data. MMBench [48] also examines LVLMs on general perception capabilities using a wide range of tasks. We also present our comparisons on ScienceQA [49], Vizwiz [22] and GQA [28] that examines certain perception capability, like knowledge and relation. Note that MMStar [8] is a vision-indispensible benchmark, which requires better visual grounding in trained LVLMs. We present our results against baseline model [46] in Tab. 5. In comparison to our baseline model LLaVA, our positional alignment approach achieves non-negligible gains across all six benchmarks, without introducing additional computation during training. Such performance gains highlight potential of Concentric Causal Attention on enhancing general visual perception capability of LVLMs.

Table 5: **Evaluation on Multiple-Choice Benchmarks**. Baseline results are reported by paper [45], except for MMStar reported by [8]. $SEED^A$, $SEED^I$ and $SEED^V$ refers to *all*, *image* and *video* evaluation, respectively. SeVa results are reported by [85].

| Method | $SEED^A$ [36] | $SEED^I$ [36] | $SEED^V$ [36] | SQA [49] | GQA [28] | VizWiz [22] | MMBench [48] | MMStar [8] | TextVQA [60] |
|---|---|---|---|---|---|---|---|---|---|
| LLaVA [45] | 58.6 | 66.1 | 37.3 | 66.8 | 62.0 | 50.0 | 64.3 | 30.0 | **58.2** |
| LLaVA w/ VCD [34] | 58.3 | 63.7 | 37.6 | 68.5 | 61.9 | 50.5 | - | **34.6** | 54.4 |
| Seva-7b-dif [85] | - | 65.8 | - | 67.5 | 60.7 | - | **65.6** | - | - |
| Seva-7b-moco [85] | - | 65.5 | - | 67.1 | 60.9 | - | 65.2 | - | - |
| CCA-LLaVA (ours) | **61.7** | **67.1** | **41.0** | **69.8** | **63.1** | **57.6** | 65.4 | 33.2 | 57.8 |

## D.2 Qualitative Comparison

We present qualitative comparison between responses generated by baseline model [46] and our trained model. We show that baseline model are more likely to hallucinate, for example, bathtub and sink in Fig. 6, knife and cup in Fig. 7. We also show case study on LLaVA-Bench [46] as illustrated in Fig. 8, where baseline model hallucinates with object hat. We also note that baseline model hallucinates in optical character recognition and numbers as in Fig. 9, where our method mitigates these issues.

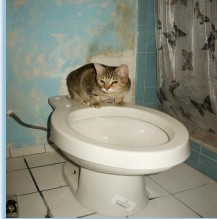

**User:** Please describe this image in detail.

**LLaVA-1.5-7B:** The image features a small cat sitting on top of a white toilet in a bathroom. The cat appears to be curiously observing its surroundings. The toilet is situated next to bathtub, and there is a shower curtain in the background. Additionally, there is a sink in the bathroom, located towards the right side of the scene.

**CCA-LLaVA-1.5-7B:** The image features a small cat sitting on the edge of a white toilet in a bathroom. The cat appears to be curiously observing its surroundings. The toilet takes up a significant portion of the scene, extending from the left to the right side of the image.

Figure 6: Qualitative comparison of open-ended generation between baseline and our method.

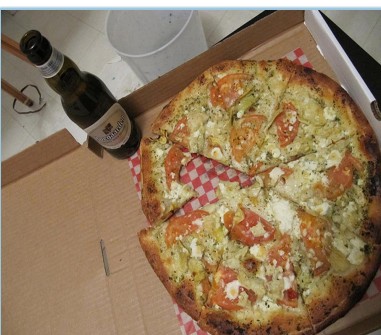

**User:** Please describe this image in detail.

**LLaVA-1.5-7B:** The image features a delicious pizza sitting in a box on dining table. The pizza is topped with tomatoes and cheese, making it an appetizing sight. Next to pizza, there is a bottle of beer, adding to the enjoyment of the meal. Addition to pizza and beer, there is knife on the table, likely used for cutting the pizza into slices. A cup can also be seen on the table, possibly containing a beverage to accompany meal.

**CCA-LLaVA-1.5-7B:** The image features a large pizza sitting on top of a cardboard box, ready to be eaten. The pizza is topped with tomatoes and cheese, making it a delicious and appetizing meal. Beside the pizza, there is a bottle of beer, adding to the enjoyment of the meal. The pizza and beer are placed on dining table, creating a cozy and inviting atmosphere.

Figure 7: Qualitative comparison of open-ended generation between baseline and our method.

# E   Computation Resources

We use 4 NVIDIA RTX A6000s to train our models. Pre-training stage takes 12 hours while instruction tuning takes roughly 83 hours.

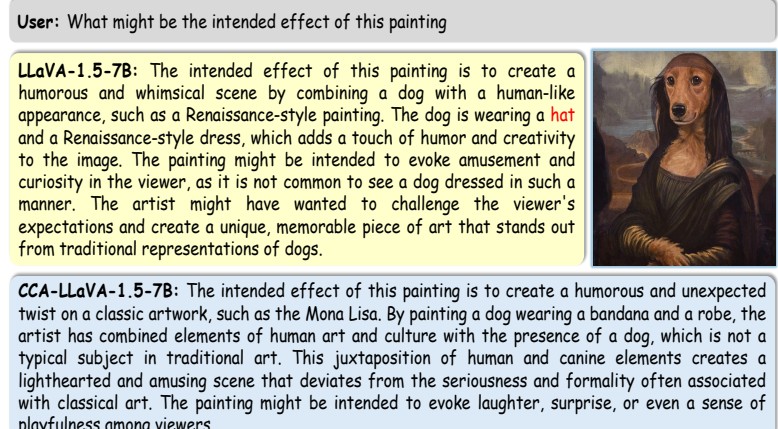

Figure 8: Case Study where question is sampled from LLaVA-Bench [46]. LLaVA hallucinates hat in its long response, while CCA answers correctly without hallucination.

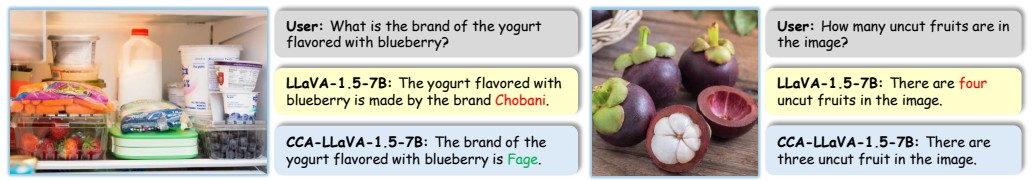

Figure 9: Case Study where question is sampled from LLaVA-Bench [46]. CCA-LLaVA outperforms LLaVA on optical character recognition (left) and numerical prediction in given cases.

