# OpenReview forum: "Mitigating Object Hallucination via Concentric Causal Attention"
_NeurIPS.cc/2024/Conference — NeurIPS 2024 poster_

### Official Review · Reviewer_5R3r · 2024-07-06

**Soundness:** 3
**Presentation:** 2
**Contribution:** 2
**Rating:** 5
**Confidence:** 5

**Summary:**

The paper attributes hallucinations in Large Vision-Language Models (LVLMs) to Rotary Positional Encoding (RoPE). It observes that LVLMs inherit a long-term decay issue from RoPE, where the inner-product of two tokens decays relative to their distance. This results in weaker visual-text interactions when the tokens are distant, leading to more frequent hallucinations if relevant visual tokens are far from the current generating token. To address this, the paper proposes a novel method called Concentric Causal Attention (CCA) to mitigate the effects of RoPE. Experiments show that CCA effectively reduces hallucinations and enhances the perception capability of LVLMs.

**Strengths:**

1. Touching the positional encoding aspect is quite fresh and particularly advantageous as it avoids the latency and multiple inference requirements seen in contrastive decoding methods such as VCD [1] and M3ID [2], which need two output probability distributions.
2. Experimental results are quite strong.

---

[1] Mitigating Object Hallucinations in Large Vision-Language Models through Visual Contrastive Decoding, CVPR 2024.
[2] Multi-Modal Hallucination Control by Visual Information Grounding, CVPR 2024.

**Weaknesses:**

1. The method is appealing as it addresses a different aspect than existing methods, but the writing can be improved in several areas. For example, the abstract should include short background information on RoPE and an explanation of the long-term decay problem, which will guide readers more friendly reading. Additionally, the captions for Figures 1 and 2 could be more concise and communicate more effectively to enhance readability. Figures 2 and 3 are difficult to understand and need clearer presentation.
2. Since long-term decay is an inherent problem of RoPE, the paper should compare RoPE with other standard positional encodings (e.g., absolute, relative, learnable positional encodings) as well as some advanced positional encodings. This would provide a more comprehensive analysis of the issue.
3. It is important to note that VCD [1] and OPERA [3] are training-free methods. Therefore, comparisons should also include more recent training-based methods.

---

[1] Mitigating Object Hallucinations in Large Vision-Language Models through Visual Contrastive Decoding, CVPR 2024.
[2] OPERA: Alleviating Hallucination in Multi-Modal Large Language Models via Over-Trust Penalty and Retrospection-Allocation, CVPR 2024.

**Questions:**

1. In Figure 1, why are the attention weights higher at the beginning and the end? If this is related to the image content itself, then Figure 1 needs to show both the image and the textual query together to determine whether the attention is well distributed with respect to the query-object relationship. If the example is averaged over 3k POPE examples, this should be clearly stated.
2. Is the method applicable to other widely used models like InstructBLIP [1] and Qwen-VL [2]? If these models do not use RoPE, the paper should mention this and discuss the implications.
3. Does the CCA method end positional encoding at the center because objects are statistically more likely to be located at the center? Is this the reason for the higher attention at the beginning and the end in Figure 1(b)? Empirical evidence supporting this assumption should be provided.

---

[1] InstructBLIP: Towards General-purpose Vision-Language Models with Instruction Tuning, NeurIPS 2023
[2] Qwen-VL: A Versatile Vision-Language Model for Understanding, Localization, Text Reading, and Beyond, Arxiv 2023

**Limitations:**

If the method is limited to LVLMs that use RoPE (e.g., LLaVA, MiniGPT), it is necessary to mention this as a limitation.

---

> ### Author Rebuttal · Authors · 2024-08-06
>
> Thank you for your meticulous reading and giving credit to our novelty and analysis of Rotary Position Encoding (RoPE) and LVLM hallucination. We appreciate you pointing out some additional references,  which we would include to make our research more complete. Please find our responses as follows.
>
> **W1: Clarification on RoPE.**
>
> **A:** Thanks for your kind advice. We will include a short background information on RoPE to indicate that, just as absolute and learnable position encoding, RoPE is also a type of position encoding, which is adopted by existing Large Language Models like LLaMA and inherited by most open-sourced LVLMs. We will also include a detailed guidance in Appendix on definition of RoPE and how it is involved in LLaMA architecture. For now we refer to [55] and lines 127-149 in our manuscript, where we present a mathematical form of RoPE and its long-term decay property for reference. For further clarification of Figure 2 in our manuscript, please find a new illustration in `Figure 4` of uploaded `pdf`. For Figure 3, please refer to `Figure 1 (right)` of uploaded `pdf` for a new illustration, where (a) to (f) corresponds to the first eight rows in Figure 3.d of our manuscript.
>
> **W2: Alternative position encoding.**
>
> **A:** Thanks for pointing this out. We kindly point out that RoPE is the default position encoding in LLaMA. Simply replacing RoPE with other position encodings is technically viable but deviates from LLaMA training scheme. We train a model with learnable position encoding and a model with relative position encoding [C] on this. Due to resource limitation, we train both models on 20k instruction data (instead of 665k) while train a new CCA model that follows the same setup for fair comparison. According to our results below, LVLM (learnable) performs much worse than LVLM (cca), while training of LVLM (relative) do not converge and not viable.
>
> |**POPE (instruction-tune 20k)**|ran||pop||adv||
> |-|-|-|-|-|-|-|
> ||acc|f1|acc|f1|acc|f1|
> |**COCO**|||||||
> |learnable|85.93|85.83|83.80|84.02|77.73|79.28|
> |ours|**88.60**|**88.33**|**85.00**|**85.20**|**81.50**|**82.35**|
> |**GQA**|||||||
> |learnable|83.33|83.94|79.67|81.08|73.23|76.50|
> |ours|**85.63**|**85.97**|**79.83**|**81.37**|**75.13**|**77.98**|
> |**A-OKVQA**|||||||
> |learnable|83.47|84.01|80.50|81.67|72.27|75.80|
> |ours|**87.77**|**87.93**|**80.73**|**82.23**|**73.87**|**77.33**|
>
> **W3: Comparison with training-based methods.**
>
> **A:** We kindly remind that we compared our method with LLaVA-RLHF in Table 1 (for POPE) and Table 2 (for CHAIR) of our manuscript, a training method that mitigates object hallucination in LVLMs. We should point out that we compare our 7B model results against those from LLaVA-RLHF-13B model and we still stands out, indicating effectiveness of proposed method. As suggested, we compare CCA with a recent training method SeVa [B], which applies DPO training on LLaVA-1.5-7B. Overall our CCA model undergoes less training compute than SeVa, as SeVa applies another DPO training stage beyond LLaVA-1.5 pre-training and fine-tuning stage, while our training strictly follows LLaVA-1.5. POPE results are listed below,
>
> |**POPE**|ran||pop||adv||
> |-|-|-|-|-|-|-|
> ||acc|f1|acc|f1|acc|f1|
> |**MSCOCO**|||||||
> |SeVa7B-MoCo|**89.43**|**88.88**|**87.23**|**86.88**|82.47|82.82|
> |ours|88.03|86.65|86.87|85.54|**85.67**|**84.42**|
> |**A-OKVQA**|||||||
> |SeVa7B-MoCo|89.96|**90.34**|84.33|85.70|75.57|79.35|
> |ours|**90.27**|89.71|**88.40**|**87.98**|**82.30**|**82.74**|
> |**GQA**|||||||
> |SeVa7B-MoCo|**89.27**|**89.73**|79.67|82.17|75.67|79.39|
> |ours|88.40|87.68|**86.47**|**85.91**|**82.20**|**82.37**|
>
> Despite less overall training compute, we highlight that our LLaVA-1.5-7B-CCA still outperforms SeVa-7B on 6 POPE evaluations. For more challenging adversarial evaluations , LLaVA-1.5-7B-CCA surpasses SeVa-7B by large margins consistently. We also compare our results with SeVa on LVLM benchmarks, where results of SeVa are taken directly from their paper. We show that our model surpasses SeVa-7B in most cases.
>
> |Model|SEED-all|SEED-img|SEED-vid|SQA|GQA|VizWiz|MMBench|MMStar|TextVQA|
> |-|-|-|-|-|-|-|-|-|-|
> |llava-1.5-7b|58.6|66.1|37.3|66.8|62.0|50.0|64.3|30.0|**58.2**|
> |vcd-llava-1.5-7b (new)|58.3|63.7|37.6|68.5|61.9|50.5|-|**34.6**|54.4|
> |seva-7b-diffu800 (new)|-|65.8|-|67.5|60.7|-|**65.6**|-|-|
> |ours|**61.7**|**67.1**|**41.0**|**69.8**|**63.1**|**57.6**|65.4|33.2|57.8|
>
> **Q1: Figure 1.**
>
> Thank you for your question. We would like to clarify that higher aggregation values at the beginning and end of Figure 1.b (manuscript) is not related to image content as we averaged over 3k COCO images to get visualization results (as detailed in Appendix B.1). This may be attributed to removal of RoPE, which leads to out-of-training-distribution and breaks pre-trained LLaMA position encoding.
>
> **Q2: Applicability to Q-formers.**
>
> Thanks for sharing this concern. Both InstructBLIP [14] and Qwen-VL [4] adopt RoPE in their language models. Nevertheless, our 2-D positional alignment strategy is designed for spatial-locality-preserved LVLMs [42,41,5], where the full image embedding from vision encoder is kept. Applying CCA to models like InstructBLIP and Qwen-VL is technically viable but not our design intentions. We will include this as a limitation of our method instead.
>
> **Q3: Empirical evidence on our concentric design.**
>
> Yes. The concentric design is motivated by that more objects are statistically located at the centre. Please find a statistical evidence collected from COCO and GQA in `Figure 3` of uploaded `pdf`. Please refer to our response for **Q1** where we clarify results from Figure 1.b (manuscript).
>
> **L1: RoPE.**
>
> Thanks for pointing this out. We admit that our method cannot apply to models with position encodings other than RoPE. We will include this in Limitation part. However, we think it is not a major drawback as most existing LVLMs use LLaMA as language backbone, where RoPE is applied as position encoding scheme.

---

> > ### Comment · Reviewer_5R3r · 2024-08-10
> >
> > First of all, I appreciate for attempting to answer all of the weaknesses I have mentioned in a careful manner. Their clarifications on RoPE, including the background information and additional illustrations, have addressed my concerns. The comparison with alternative position encodings and training methods, particularly the empirical results the authors provided, strengthens the validity of the approach. I appreciate the clarification on Figure 1 and the explanation of the concentric design, both of which are now much clearer to me. Including the limitations regarding RoPE and the applicability to Q-formers demonstrates transparency and further enhances the quality of this work. Overall, the rebuttal has significantly improved my understanding, and I am raising my score accordingly.

---

> > > ### Author Response · Authors · 2024-08-11
> > >
> > > Thank you for the follow-up comment. We sincerely appreciate your valuable feedback and recognition of our rebuttal efforts. The additional analyses and experiments you suggested have helped us strengthen our paper and solidify our approach. If you have any further question or comment, we are eager to address them.

---

### Official Review · Reviewer_5Msc · 2024-07-11

**Soundness:** 2
**Presentation:** 2
**Contribution:** 2
**Rating:** 5
**Confidence:** 4

**Summary:**

The paper shows that object hallucination in LVLMs is linked to the commonly adopted Rotary Position Encoding (RoPE) strategies. The long-term decay in RoPE causes hallucinations when important visual tokens are distant from visual instructions. To address this, the authors propose the Concentric Causal Attention strategy to reduce the distance between these tokens. However, experimental results of the proposed method is not promising on several benchmark datasets.

**Strengths:**

The analysis of the long-term decay in RoPE and its impact on hallucinations in LVLMs is novel.

**Weaknesses:**

- The method section is quite short, and it seems not covering a comprehensive solution to address LVLM hallucination problem. It lacks justification on the proposed scanning method. Specifically, the concentric causal masking section is not clear.

- Performance of the proposed method is not promising on several benchmark datasets. For example, the proposed method sometimes achieves the best results only when combined with other state-of-the-art methods. It might provide more insight if there is an analysis on when and why the proposed CCA alone will work or fail (Table 2).

- The detailed description of the method with the figure is not clear. Please see the questions below.

**Questions:**

- Please clarify what do the orange and yellow colors represent in Figure 3(d)? The explanation of casual masking in Fig.3d is not clear: need guidance on interpreting the concentric causal masking in Figure 3.

- What is the distribution of the aggregated correct responses in Figure 2 with the proposed concentric causal attention? Does the distribution different from the raster scan and reverse raster scan?

- The paper compared with raster-scan as the baseline. But how do you justify that the concentric positional assignment in Figure 3 is the best solution (or is good enough)? Did you consider other scan designs, for example, the diagonal-scan, zig-zag scan design?

**Limitations:**

Yes, the author adequately addressed the limitations and potential negative societal impact of their work.

---

> ### Author Rebuttal · Authors · 2024-08-06
>
> Thank you for your valuable insights. Please see our responses to your questions below.
>
> **W1: Alternative scanning method.**
>
> **A:**: We justify the design of our method by providing new comparative studies for different position encoding schemes and alternative scanning methods. We first compare CCA with learnable position encoding. Due to resource limitations, we train all positional alignment approaches on 558k pre-training data and only 20k instruction data, including a new CCA model with same setup for fair comparison. We conduct evaluation on POPE and CHAIR benchmarks and the results are shown in tables below. The resulting models with learnable position encoding perform worse than our design.
>
> |**POPE (instruction-tune 20k)**|ran||pop||adv||
> |-|-|-|-|-|-|-|
> ||acc|f1|acc|f1|acc|f1|
> |**MSCOCO**|||||||
> |learnable|85.93|85.83|83.80|84.02|77.73|79.28|
> |ours|**88.60**|**88.33**|**85.00**|**85.20**|**81.50**|**82.35**|
> |**GQA**|||||||
> |learnable|83.33|83.94|79.67|81.08|73.23|76.50|
> |ours|**85.63**|**85.97**|**79.83**|**81.37**|**75.13**|**77.98**|
> |**A-OKVQA**|||||||
> |learnable|83.47|84.01|80.50|81.67|72.27|75.80|
> |ours|**87.77**|**87.93**|**80.73**|**82.23**|**73.87**|**77.33**|
>
> We also compare alternative scanning designs. We reverse the scanning order of original CCA and start from visual tokens at the center of image and end at periphery, illustrated in `Figure 2 (right)` of uploaded `pdf`. As kindly suggested in your question 3, we implemented diagonal scan and provide evaluation results for new scanning designs in a table below. Our original CCA scanning method demonstrated overall show better performance over other design choices. Based on these ablation studies, CCA is chosen as our final method for hallucination mitigation.
>
> |**POPE**|ran||pop||adv||
> |-|-|-|-|-|-|-|
> ||acc|f1|acc|f1|acc|f1|
> |**MSCOCO**|||||||
> |CCA-r (new)|87.43|85.90|86.33|84.85|85.17|83.77|
> |diagonal-lora (new)|**88.10**|**86.72**|**87.17**|**85.83**|**85.70**|**84.46**|
> |CCA|88.03|86.65|86.87|85.54|85.67|84.42|
> |CCA-lora (new)|88.03|86.68|87.13|85.82|85.50|84.30|
> |**GQA**|||||||
> |CCA-r (new)|88.63|87.99|83.43|83.41|81.83|82.09|
> |diagonal-lora (new)|89.07|88.38|85.80|85.41|**82.70**|**82.77**|
> |CCA|**90.27**|**89.71**|**88.40**|**87.98**|82.30|82.74|
> |CCA-lora (new)|89.30|88.70|85.40|85.22|82.47|**82.77**|
> |**A-OKVQA**|||||||
> |CCA-r (new)|89.70|89.12|86.90|86.56|81.20|81.78|
> |diagonal-lora (new)|90.03|89.48|**87.93**|87.53|82.10|82.55|
> |CCA|88.40|87.68|86.47|85.91|**82.20**|82.37|
> |CCA-lora (new)|**90.33**|**89.88**|87.83|**87.59**|82.13|**82.70**|
>
> |**CHAIR**|512||||64||||
> |-|-|-|-|-|-|-|-|-|
> ||c_s (↓)|c_i (↓)|rec|len|c_s (↓)|c_i (↓)|rec|len|
> |diagonal-lora (new)|49.0|13.6|79.6|93.1|**15.6**|5.2|64.9|52.6|
> |CCA-r (new)|50.0|18.1|**85.7**|96.4|18.0|5.5|66.1|54.7|
> |CCA|**43.0**|**11.5**|80.4|96.6|18.2|5.4|**66.7**|54.5|
> |CCA-lora (new)|45.0|12.4|80.8|93.3|17.2|**5.2**|65.4|52.7|
>
> For presentation of concentric causal masking, please find new illustrations in `Figure 1` of uploaded `pdf`.
>
> **W2: Performance on CHAIR.**
>
> **A:** Thanks for pointing this out. We kindly remind that our trained model achieved the best results (lowest CHAIR scores) when applying CCA alone, as shown in the table below (summarized from manuscript Table 2). We point out that the LLaVA-RLHF model for benchmarking uses stronger Vicuna-13B as its language backbone and involves additional direct preference optimization stage in their training, whereas our model undergoes only two training stages (pre-training and supervised fine-tuning). Despite smaller 7B language backbone and less training compute, our model still outperforms the LLaVA-RLHF-13B model on most metrics.
>
> |CHAIR||512||||64||||
> |-|-|-|-|-|-|-|-|-|-|
> |||c_s (↓)|c_i (↓)|rec|len|c_s (↓)|c_i (↓)|rec|len|
> |LLaVA-RLHF-13B-v1.5-336|greedy|43.6|**10.5**|78.0|117.9|19.6|5.4|64.9|54.0|
> |OPERA|beam|46.8|13.4|79.6|93.2|17.8|5.9|64.3|53.0|
> |ours|greedy|**43.0**|11.5|**80.4**|96.6|18.2|5.4|**66.7**|54.5|
> |ours|beam|48.6|13.4|79.9|94.2|**16.0**|**5.3**|64.8|52.7|
>
> We also point out that our method benefits POPE as well (manuscript Table 1), indicating good compatibility of our method on both open-ended generation and yes-no tasks. Moreover, we highlight that our method also benefits general perception tasks, where approaches that exclusively address object hallucination cannot always bring performance gain. Please refer to Table 5 in Appendix C.2, where we show that our trained model LLaVA-1.5-7B-CCA surpass LLaVA-1.5-7B over multiple LVLM benchmarks consistently.
>
> **Q1: Clarification on concentric causal masking.**
>
> **A:** In Figure 3 (d), we use different colours to highlight visual tokens with different positions. For a 2-D organization of visual tokens with shape of 6x6, our CCA leads to 3 positions in visual tokens. Please refer to our new illustrations in `Figure 1` of uploaded `pdf`, where query tokens, key tokens and masked tokens (tokens not involved in self-attention computation) are highlighted. Our CCA follows the same causal modelling rule in LLaMA Figure 1, where query tokens with larger position values attend to key tokens with smaller or equal position values. The difference is that we use a 2-D positional organization for visual tokens, which is a novel and effective attempt among existing LVLM hallucination studies.
>
> **Q2: Aggregated correct responses of concentric causal attention.**
>
> **A:** Thanks for mentioning this. We visualize aggregated correct responses in `Figure 5` of uploaded `pdf` with the proposed concentric causal attention. The resulting distribution is largely different from that from raster scan in Figure 2.a and reverse raster scan in Figure 2.b of our manuscript, showing 2-D and symmetrical distribution, which aligns with our concentric causal design.
>
> **Q3: Alternative position encodings and scanning.**
>
> **A:** Please refer to our response in **W1**.

---

> ### Comment · Reviewer_5Msc · 2024-08-13
>
> I appreciate the authors' additional experimental results and clarifications. However, there appears to be inconsistent performance across datasets, such as diagonal-lora outperforming the proposed CCA/CCA-lora in MSCOCO and on the adv part of GQA. The authors have suggested that the performance drop of CCA-r supports the assumption that most image content is concentrated in central regions, aligning with their proposed design. Based on this suggestion, how can we interpret the better performance of the diagonal approach? A more in-depth discussion of these inconsistencies would provide greater insights into the effectiveness and the limitation of the proposed CCA strategy. Nonetheless, I believe the author have provided in-depth answers to many asked questions; I have raised my scores to 5.

---

> > ### Author Response · Authors · 2024-08-14
> >
> > Thank you for recognising the value of our rebuttal and raising your score. We appreciate your effort in reading through our reply and pointing out your new concern. We will continue to explore alternative position encoding and scanning methods as suggested to improve our study.

---

### Official Review · Reviewer_1NX2 · 2024-07-12

**Soundness:** 3
**Presentation:** 2
**Contribution:** 3
**Rating:** 6
**Confidence:** 4

**Summary:**

This paper analyze the long-term dependency between text token and visual token in LVLMs from a novel positional encoding perspective by replace the RoPE method. The analysis shows that RoPE introduce clear long-term decay regarding the attention scores. The authors propose a novel concentric causal attention (CCA) including visual token re-organization and concentric causal masking to keep 2-D spatial locality while shortens visual-instruction distance. Experimental results show CCA can reduce hallucination on both discriminative and generative benchmarks while keep comparable performance on general benchmarks.

**Strengths:**

- The motivation of exploring source of hallucination from a long-term decay perspective is clear.
- The paper is well organized and clear.
- The proposed CCA method seems effective and easy-to-implement.

**Weaknesses:**

- The experimental setup lacks rigor. The VCD results are sourced from the original paper, which is based on LLaVA 1.5. However, the baseline provided by the authors utilizes the pre-training scheme of LLaVA 1.0 along with the projection module and instruction fine-tuning data from LLaVA 1.5. This discrepancy raises concerns about the comparability of the results. Additionally, it is unclear why Table 3 does not include results for opera and why Table 4 omits the VCD results.

**Questions:**

- In Figure 1  (b), would the removal of the RoPE positional encoding makes the self-attention process out of distribution? Since the model is pre-trained with RoPE applied. How would this affect the quantitative results?
- The explanation of concentric causal masking in Figure 3 and the corresponding paragraph is somewhat unclear. Could you clarify why the attention masks for the first seven rows are identical? A more comprehensive and detail introduction helps a lot.

**Limitations:**

authors adequately addressed the limitations

---

> ### Author Rebuttal · Authors · 2024-08-06
>
> Thanks for your detailed and insightful suggestions. Please find our responses as follows.
>
> **W1-a: Pretraining setup.**
>
> **A:** Thanks for mentioning this concern. We would share that it is a typo in line 227 where we claim we use CC-595K dataset [42] for pre-training stage. In fact, our pre-training experiments follow LLaVA 1.5 [41] and use a 558K dataset for pre-training. The provided baseline results in Table 1 of our manuscript is for LLaVA 1.5, which are sourced from VCD paper [30]. We will release our model and source code for the community to reproduce our results.
>
> **W1-b: Manuscript Table 3 and Table 4.**
>
> **A:** Thanks for pointing this out. Please find new quantitative results of Table 3 and 4 below. We use their official codes and get these results.
>
> |MME | Existence | Count   | Position | Color  | Total  |
> |-|-|-|-|-|-|
> | baseline          | 175.67    | 124.67  | 114.00   | 151.00 | 565.33 |
> | OPERA (new)| 180.67    | 133.33  | 123.33   | 155.00 | 592.33 |
> | VCD               | 184.66    | 138.33  | **128.67** | 153.00 | 604.66 |
> | ours              | **190.00** | **148.33** | 128.33   | **175.00** | **641.66** |
>
> |LLaVA-Bench | Complex | Detail | Conv  |
> |-|-|-|-|
> | baseline          | 65.8    | 51.2   | 54.6  |
> | OPERA             | 66.4    | **56.9**   | 44.0  |
> | VCD (new result)  | **69.6**    | 51.8   | 57.3  |
> | ours              | 66.1 | 53.9 | **69.4**|
>
> **Q1: Removal of RoPE from LLaVA.**
>
> **A:** Yes, removing RoPE largely diverges from LLaMA pre-training and leads to nonsense outputs. Our earlier studies showed that LVLMs without RoPE will no longer follow instructions. Take POPE [37] questions as an example. The LLaVA-v1.5-7B pretrained without RoPE fails to answer yes or no as illustrated below.
> > USER: Is there a scissors in the image?
> >
> > LLaVA-1.5-7B: No.
> >
> > LLaVA-1.5-7B w/o RoPE: the bear.\n\n\n\n\n\n\n\n
>
> Since the LLaVA-1.5-7B w/o RoPE will output neither yes nor no, the quantitative accuracy on POPE should be 0.00. Though the model w/o RoPE in Figure 1.b of our manuscript generates non-sense text outputs, the information flows from visual to text tokens are more evenly distributed. This highlights the long-term decay in RoPE in Figure 1.c, the root cause of information aggregating at image tokens that are closer to text tokens.
>
> **Q2: Clarifications on concentric causal masking.**
>
> Thanks for pointing this out. Please find a new illustration in `Figure 1` of uploaded `pdf` that clarifies the proposed concentric causal masking, where query tokens, key tokens attended by query tokens and key tokens not attended by query tokens (tokens not involved in self-attention computation) are colored. Consistent with our manuscript Figure 3, we take 6x6 visual token organization as an example. The design follows the same causal modeling rule of LLaVA (presented in `Figure 1 (left)` of uploaded `pdf`), where query tokens with larger position values attend to key tokens with smaller or equal position values. The first seven rows of manuscript Figure 3 (d) share the same attention masks, corresponding to `Figure 1 (right)` (a) to (g) of uploaded `pdf`, where the position indices of key tokens attended by query tokens are exactly the same.

---

> ### Comment · Reviewer_1NX2 · 2024-08-13
>
> The responses address my concerns and the provided illustration figures are great. I am curious about the attention distribution of model trained with CCA and considering raise my score.

---

> > ### Author Response · Authors · 2024-08-14
> >
> > Thank you for going through our rebuttal. We are glad that it addressed your concerns. For your follow-up question, we have prepared the new attention distribution visualisation. Unfortunately we are not allowed to provide the image through any link during author-reviewer discussion phase, according to author guideline. Instead, we present it here in tabular format for your reference. Alternating rings are bolded for better presentation.
> >
> > The table below is in 24x24, which is the suggested distribution from our proposed LLaVA-1.5-7B-CCA. As presented in this table, attention values gradually increases when positions move from periphery to center. The highest values can be found in the central position. Meanwhile, they also show a 2-D concentric distribution, with each ring having similar values. This clearly aligns with our CCA design. We will include visualisation in revision of our manuscript to better support our approach.
> > |||||||||||||||||||||||||
> > |-|-|-|-|-|-|-|-|-|-|-|-|-|-|-|-|-|-|-|-|-|-|-|-|
> > |0.02|0.07|0.29|0.13|0.20|0.50|0.57|0.56|0.13|0.18|0.34|0.33|0.34|0.41|0.24|0.14|0.44|0.16|0.05|0.14|0.13|0.16|0.12|0.07|
> > |0.07|**0.07**|**0.18**|**0.18**|**0.20**|**0.43**|**0.37**|**0.57**|**0.26**|**0.24**|**0.41**|**0.21**|**0.20**|**0.21**|**0.30**|**0.19**|**0.18**|**0.14**|**0.21**|**0.16**|**0.16**|**0.18**|**0.23**|0.00|
> > |0.06|**0.16**|0.23|0.21|0.29|0.25|0.34|0.25|0.34|0.25|0.25|0.24|0.24|0.27|0.31|0.24|0.21|0.18|0.19|0.18|0.18|0.21|**0.18**|0.13|
> > |0.15|**0.15**|0.19|**0.25**|**0.26**|**0.28**|**0.35**|**0.27**|**0.29**|**0.28**|**0.30**|**0.27**|**0.30**|**0.31**|**0.31**|**0.27**|**0.26**|**0.28**|**0.25**|**0.23**|**0.22**|0.22|**0.18**|0.19|
> > |0.11|**0.14**|0.18|**0.22**|0.28|0.28|0.31|0.34|0.31|0.31|0.31|0.33|0.32|0.34|0.31|0.31|0.31|0.30|0.28|0.27|**0.22**|0.20|**0.16**|0.11|
> > |0.05|**0.11**|0.16|**0.21**|0.27|**0.34**|**0.35**|**0.35**|**0.38**|**0.36**|**0.38**|**0.37**|**0.38**|**0.40**|**0.39**|**0.38**|**0.37**|**0.36**|**0.32**|0.27|**0.22**|0.19|**0.16**|0.14|
> > |0.09|**0.11**|0.22|**0.24**|0.27|**0.32**|0.40|0.42|0.43|0.45|0.44|0.46|0.46|0.48|0.45|0.45|0.47|0.43|**0.37**|0.30|**0.26**|0.21|**0.21**|0.15|
> > |0.06|**0.09**|0.21|**0.27**|0.29|**0.34**|0.41|**0.49**|**0.49**|**0.48**|**0.51**|**0.51**|**0.52**|**0.51**|**0.53**|**0.51**|**0.51**|0.43|**0.39**|0.32|**0.29**|0.24|**0.19**|0.15|
> > |0.15|**0.11**|0.22|**0.26**|0.31|**0.38**|0.44|**0.51**|0.57|0.59|0.60|0.58|0.62|0.62|0.62|0.63|**0.55**|0.49|**0.42**|0.33|**0.29**|0.24|**0.20**|0.15|
> > |0.07|**0.22**|0.23|**0.28**|0.33|**0.40**|0.45|**0.50**|0.61|**0.67**|**0.72**|**0.70**|**0.68**|**0.72**|**0.74**|0.63|**0.56**|0.48|**0.41**|0.36|**0.29**|0.25|**0.20**|0.20|
> > |0.09|**0.18**|0.27|**0.31**|0.36|**0.39**|0.45|**0.54**|0.61|**0.72**|0.83|0.85|0.82|0.84|**0.73**|0.66|**0.57**|0.49|**0.43**|0.38|**0.32**|0.28|**0.23**|0.19|
> > |0.11|**0.19**|0.23|**0.31**|0.37|**0.42**|0.47|**0.56**|0.64|**0.73**|0.84|**0.98**|**1.00**|0.85|**0.76**|0.67|**0.59**|0.54|**0.44**|0.38|**0.35**|0.30|**0.23**|0.23|
> > |0.18|**0.21**|0.29|**0.32**|0.37|**0.42**|0.50|**0.56**|0.64|**0.75**|0.88|**1.00**|**1.00**|0.86|**0.77**|0.67|**0.61**|0.55|**0.46**|0.41|**0.38**|0.32|**0.24**|0.24|
> > |0.18|**0.22**|0.26|**0.31**|0.37|**0.46**|0.52|**0.57**|0.66|**0.76**|0.88|0.88|0.90|0.89|**0.78**|0.69|**0.61**|0.55|**0.48**|0.42|**0.36**|0.32|**0.27**|0.20|
> > |0.16|**0.20**|0.24|**0.30**|0.36|**0.44**|0.51|**0.57**|0.66|**0.76**|**0.76**|**0.78**|**0.77**|**0.79**|**0.77**|0.70|**0.60**|0.53|**0.47**|0.41|**0.35**|0.30|**0.24**|0.26|
> > |0.15|**0.20**|0.28|**0.31**|0.37|**0.41**|0.52|**0.58**|0.67|0.68|0.65|0.68|0.68|0.71|0.68|0.71|**0.61**|0.52|**0.50**|0.39|**0.35**|0.29|**0.27**|0.18|
> > |0.25|**0.23**|0.28|**0.32**|0.37|**0.42**|0.53|**0.60**|**0.62**|**0.61**|**0.61**|**0.61**|**0.63**|**0.63**|**0.62**|**0.62**|**0.63**|0.54|**0.46**|0.38|**0.34**|0.27|**0.26**|0.22|
> > |0.11|**0.25**|0.26|**0.35**|0.39|**0.45**|0.51|0.54|0.54|0.56|0.57|0.57|0.56|0.57|0.57|0.57|0.55|0.52|**0.43**|0.39|**0.30**|0.25|**0.23**|0.19|
> > |0.13|**0.18**|0.25|**0.31**|0.38|**0.44**|**0.47**|**0.50**|**0.48**|**0.49**|**0.49**|**0.48**|**0.49**|**0.50**|**0.49**|**0.47**|**0.46**|**0.43**|**0.42**|0.35|**0.34**|0.24|**0.24**|0.19|
> > |0.13|**0.18**|0.24|**0.26**|0.38|0.41|0.43|0.44|0.40|0.40|0.41|0.41|0.39|0.40|0.41|0.39|0.39|0.36|0.34|0.33|**0.27**|0.23|**0.19**|0.25|
> > |0.07|**0.22**|0.21|**0.26**|**0.34**|**0.41**|**0.47**|**0.46**|**0.45**|**0.43**|**0.46**|**0.33**|**0.32**|**0.33**|**0.37**|**0.31**|**0.31**|**0.31**|**0.32**|**0.25**|**0.24**|0.23|**0.28**|0.17|
> > |0.07|**0.30**|0.26|0.27|0.34|0.41|0.30|0.46|0.27|0.47|0.57|0.52|0.27|0.30|0.54|0.27|0.29|0.41|0.22|0.24|0.23|0.26|**0.30**|0.16|
> > |0.05|**0.12**|**0.20**|**0.19**|**0.23**|**0.38**|**0.28**|**0.46**|**0.55**|**0.35**|**0.37**|**0.26**|**0.23**|**0.32**|**0.28**|**0.20**|**0.20**|**0.18**|**0.14**|**0.18**|**0.17**|**0.18**|**0.11**|0.04|
> > |0.02|0.11|0.10|0.07|0.13|0.13|0.20|0.59|0.16|0.19|0.17|0.17|0.15|0.18|0.23|0.16|0.13|0.15|0.12|0.13|0.06|0.12|0.16|0.11|

---

> > > ### Comment · Reviewer_1NX2 · 2024-08-14
> > >
> > > What is the query for this attention map result? If I understood it correctly, results of text tokens and image tokens are supposed to be different systematically.

---

> > > > ### Author Response · Authors · 2024-08-14
> > > >
> > > > Thank you for your prompt reply. Similar to Figure 1 (b) and (c) in our manuscript, we apply the same experiment setting to obtain these values, except that we use our trained LLaVA-1.5-7b-CCA model. The text-to-vision information flows are only for image tokens (24x24, 576 in total), so the queries and keys for getting this result are text and image tokens, respectively. The given result is structured in 2-D format for demonstration purpose and does not include self-attentions among text tokens or self-attentions among image tokens. Please refer to Appendix B.1 where we elaborated how we got these results. We are happy to discuss further if more clarifications are needed.

---

> > > > > ### Comment · Reviewer_1NX2 · 2024-08-14
> > > > >
> > > > > I raise my score to 6.

---

### Official Review · Reviewer_9xbc · 2024-07-13

**Soundness:** 4
**Presentation:** 3
**Contribution:** 4
**Rating:** 6
**Confidence:** 5

**Summary:**

This paper explores how current LVLM's hallucination appears through analyzing the impact of RoPE long-term decay on vision information  attenuation during flow. It gives clear visualization results and theoretical evidence to prove the basic point, revealing that the causal attention mask and RoPE embbeding is not appropriate for non-text-modal input (e.g., vision tokens). Based on these, the authors argue that LVLM's hallucinations are mainly attributed by the RoPE long-term decay and the mismatch between vision tokens and causal attentions. To this end, the authors propose vision token re-organization and concentric causal attention masking to alleviate the hallucination. Experiments demonstrate the promising performance of the proposed methods.

**Strengths:**

1. This paper gives a good explanation for the relationship between hallucination and long-term decay in LVLMs.
2. The proposed method is well motivated and technically sound.

**Weaknesses:**

1. When introducing the long-term decay and the attenuation of information flow, it would be better to cite some references like OPERA (some of the early works that claim the relationship between hallucination and long-term decay).
2. One suggestion for the experiment on Figure 2: Although the authors calculate and visualize the results on thousands of samples, it would be better to reverse the order of vision patches (before CLIP-ViT) and conduct the same experiment again. We should rule out the possibility that most of correct answers are naturally located at the lower region of images.
3. Both the proposed method relies on the hypothesis that the most of main contents of images will be located at the centric region. It generally make sense, but not enough to be an accurate solution.
4. The experiments are somehow insufficient. It would be better to add some ablation studies and evaluation on LVLM benchmarks. MME hallucination split and LLaVA Bench are not enough. It would be better to add results on MMBench, Seed Bench, TextVQA, etc.

**Questions:**

See the Weaknesses.

---

> ### Author Rebuttal · Authors · 2024-08-06
>
> Thanks for your detailed and thorough suggestions. Please find our replies as below.
>
> **W1: CCA and OPERA [23].**
>
> **A:** We ground our design on analysis of information flow in LLaVA model. This shares commonalities with OPERA which analyzes information flow in LVLM autoregressive decoding. Thank you for pointing out and we would highlight this in introduction of our manuscript as suggested. Different from OPERA that discovers co-occurrences of aggregation pattern and object hallucination, our CCA explores relations between Rotary Position Encoding and object hallucination. We further refer to quantitative results in Table 2 and 4 of our manuscript, where proposed CCA surpasses OPERA in CHAIR and LLaVA-Bench evaluations.
>
> **W2: Figure 2 experiments.**
>
> **A:** It is a valid concern to negate the impact of imbalanced object distribution in different image regions. We kindly let you know that we already addressed this in the paper (lines 167-169) by cropping objects and pasting them on blank images (initialized with ImageNet mean pixel values) at different spatial positions to create synthesized images. Please find a new illustration for this in `Figure 4` of uploaded `pdf`, where 16 (4 by 4) pasting options are given. In our experiments we use 144 (12 by 12) pasting options. Figure 2 of our manuscript is obtained by testing models on these synthesized images which have even distribution of objects at different regions.
>
> **W3: Data statistics and our concentric design.**
>
> **A:** Yes, our concentric design assumes that most image contents are located around the centric image region. We validate this assumption from two perspectives.
>
> Firstly, we perform statistical analysis on large amount of natural images (82,081 images and 604,907 annotations from COCO train 2014, and 10,696 images and 174,304 annotations from GQA). Specifically, we count total number of objects in 9 spatial locations (top_left, top_mid, top_right, mid_left, center, mid_right, bottom_left, bottom_mid, bottom_right, respectively). The statistical results in `Figure 3` of uploaded `pdf` show that more objects are located in centre, which aligns with our design.
>
> Secondly, we point out that for our model, positions start from periphery of 2-D visual tokens and ends in centre. For comparison, we implement another LVLM (which we name as CCA-r), where positions start from centre of 2-D visual tokens and ends in periphery. Please find an illustration of CCA-r in `Figure 2` of uploaded `pdf`, where left refers to our CCA method and right refers to CCA-r. Query tokens, key tokens for self-attention calculations are highlighted with colors.
>
> The tables below provides quantitative experiments on POPE and CHAIR, showing that changing from CCA to CCA-r positional alignment causes performance drop in most evaluations. For GQA popular evaluation, accuracy drops from 88.40 to 83.43. For CHAIR evaluation, c_s drops from 43.0 to 50.0. These validate the assumption that most image contents are located around centric image regions, which aligns with our design.
>
> |**POPE**|ran||pop||adv||
> | - | - | - | - | - | - | - |
> ||acc|f1|acc|f1|acc|f1|
> |**MSCOCO**||||             |               |               |
> | CCA-r (new) | 87.43    | 85.90       | 86.33       | 84.85       | 85.17         | 83.77         |
> | ours           | **88.03**     | **86.65**   | **86.87**   | **85.54**   | **85.67**     | **84.42**     |
> | **GQA**        |               |             |             |             |               |               |
> | CCA-r (new) | 88.63    | 87.99       | 83.43       | 83.41       | 81.83         | 82.09         |
> | ours           | **90.27**     | **89.71**   | **88.40**   | **87.98**   | **82.30**     | **82.74**     |
> | **A-OKVQA**    |               |             |             |             |               |               |
> | CCA-r (new) | **89.70**   | **89.12**       | **86.90**       | **86.56**       | 81.20         | 81.78         |
> | ours           | 88.40         | 87.68       | 86.47       | 85.91       | **82.20**         | **82.37**         |
>
> |**CHAIR**|512||||64||||
> |-|-|-|-|-|-|-|-|-|
> ||c_s (↓)|c_i (↓)|rec|len|c_s (↓)|c_i (↓)|rec|len|
> |CCA-r (new)|50.0|18.1|**85.7**|96.4|**18.0**|5.5|66.1|54.7|
> |CCA|**43.0**|**11.5**|80.4|96.6|18.2|**5.4**|**66.7**|54.5|
>
> **W4: More evaluations on LVLM benchmarks.**
>
> **A:** Please refer to Table 5 in Appendix C.2 where we included more evaluations on LVLM benchmarks. As suggested, we also add TextVQA [A] for your reference. We also compare general perception capabilities on these benchmarks against two hallucination-mitigating methods [30, B], where SeVa [B] is a recent method that explores unsupervised preference alignment in LVLMs. We point out that SeVa trains their models with another Direct-Preference-Optimization stage on top of LLaVA-1.5 [41] whereas our CCA does not involve any new training stages and strictly follows LLaVA-1.5 training scheme. Our model LLaVA-1.5-7B-CCA outperforms SeVa-7B and VCD on most LVLM benchmarks.
>
> | Model                          | SEED-all | SEED-img | SEED-vid | SQA | GQA  | VizWiz | MMBench | MMStar | TextVQA |
> | - | - | - | - | - | - | - | - | - | - |
> | llava-1.5-7b                   | 58.6     | 66.1       | 37.3       | 66.8       | 62.0 | 50.0   | 64.3    | 30.0   | **58.2**    |
> | vcd-llava-1.5-7b (new) | 58.3     | 63.7       | 37.6       | 68.5       | 61.9 | 50.5   | -       | **34.6**   | 54.4    |
> | seva-7b-diffu500 (new) | -        | 65.8       | -          | 67.4       | 61.1 | -      | 64.7    | -      | -       |
> | seva-7b-diffu800 (new) | -        | 65.8       | -          | 67.5       | 60.7 | -      | **65.6**    | -      | -       |
> | seva-7b-moco (new)     | -        | 65.5       | -          | 67.1       | 60.9 | -      | 65.2    | -      | -       |
> | llava-1.5-7b-cca (ours)        | **61.7**     | **67.1**       | **41.0**       | **69.8**       | **63.1** | **57.6**   | 65.4    | 33.2   | 57.8    |

---

> > ### Comment · Reviewer_9xbc · 2024-08-13
> >
> > Thanks for the reply. It addresses the most of my concerns. I hope the authors can add these results in the revision.

---

> > > ### Author Response · Authors · 2024-08-13
> > >
> > > Thank you for your thorough and insightful review of our paper. We are happy that our rebuttal has addressed most of your concerns. We will include rebuttal experiments in our revision as suggested. If any residual concerns remain, we would be glad to discuss further. If no concerns remain, we would appreciate it if you could raise your score.

---

### Author Rebuttal · Authors · 2024-08-06

We sincerely appreciate reviewers `9xbc` and `1NX2` for acknowledging clear motivation behind our work, and reviewers `5Msc` and `5R3r` for recognizing novelty of our study, along with thoughtful and kind suggestions for improving our paper. Please find new figures in attached `pdf`. New figures in our rebuttal texts are highlighted with `Figure x`. We also include new references here for all reviewers.

> [A] Towards VQA Models That Can Read.
>
> [B] Self-Supervised Visual Preference Alignment.
>
> [C] Self-Attention with Relative Position Representations.

---

### Decision · Program_Chairs · 2024-09-25

**Decision:**

Accept (poster)

**Comment:**

Summary:
This paper investigates how hallucinations in large vision-language models (LVLMs) are influenced by the RoPE long-term decay and its effect on vision information attenuation. The authors provide both visualizations and theoretical evidence showing that causal attention masks and RoPE embeddings are not well-suited for non-textual inputs like vision tokens. To address these issues, the paper proposes vision token re-organization and concentric causal attention masking, demonstrating improved performance in experiments.

Strengths:
The paper offers a strong explanation of the relationship between hallucinations and long-term decay in LVLMs, supported by both theoretical and empirical evidence.

All reviewers agree to accept the submission.